collaborative care; integrated care; LMIC; implementation; mental health disorders; mhGAP; substance use disorders; health care delivery; mental health integration; outpatient mental health care; depression

**Author for correspondence:**
Jessica Whitfield,
Email: jwhitfi@uw.edu

# Successful ingredients of effective Collaborative Care programs in low- and middle-income countries: A rapid review

Jessica Whitfield[1,2] , Shanise Owens[3], Amritha Bhat[1], Bradford Felker[1] ,
Teresa Jewell[4] and Lydia Chwastiak[1,2,5]

[1]Department of Psychiatry and Behavioral Sciences, University of Washington School of Medicine, Seattle, WA, USA; [2]Advancing Integrated Mental Health Solutions (AIMS) Center, University of Washington, Seattle, WA, USA; [3]Department of Health Systems and Population Health, University of Washington School of Public Health, Seattle, WA, USA; [4]University of Washington Health Sciences Library, University of Washington, Seattle, WA, USA and [5]Department of Global Health, University of Washington School of Public Health, Seattle, WA, USA

## Abstract

Integrating mental health care in primary healthcare settings is a compelling strategy to address the mental health treatment gap in low- and middle-income countries (LMICs). Collaborative Care is the integrated care model with the most evidence supporting its effectiveness, but most research has been conducted in high-income countries. Efforts to implement this complex multi-component model at scale in LMICs will be enhanced by understanding the model components that have been effective in LMIC settings. Following Cochrane Rapid Reviews Methods Group recommendations, we conducted a rapid review to identify studies of the effectiveness of Collaborative Care for priority adult mental disorders of mhGAP (mood and anxiety disorders, psychosis, substance use disorders and epilepsy) in outpatient medical settings in LMICs. Article screening and data extraction were performed using Covidence software. Data extraction by two authors utilized a checklist of key components of effective interventions. Information was aggregated to examine how frequently the components were applied. Our search yielded 25 articles describing 20 Collaborative Care models that treated depression, anxiety, schizophrenia, alcohol use disorder or epilepsy in nine different LMICs. Fourteen of these models demonstrated statistically significantly improved clinical outcomes compared to comparison groups. Successful models shared key structural and process-of-care elements: a multi-disciplinary care team with structured communication; standardized protocols for evidence-based treatments; systematic identification of mental disorders, and a stepped-care approach to treatment intensification. There was substantial heterogeneity across studies with respect to the specifics of model components, and clear evidence of the importance of tailoring the model to the local context. This review provides evidence that Collaborative Care is effective across a range of mental disorders in LMICs. More work is needed to demonstrate population-level and longer-term outcomes, and to identify strategies that will support successful and sustained implementation in routine clinical settings.

## Impact statement

Integrating mental health care into outpatient medical settings, such as primary care, HIV and diabetes clinics, is an effective strategy to address the tremendous global mental health treatment gap. Collaborative Care is an integrated care model with the largest evidence base supporting its effectiveness for a range of mental disorders. It involves multiple components: team-based care, structured communication between providers, tracking patient progress systematically and evidence-based treatments like pharmacotherapy and behavioral interventions. However, most research around Collaborative Care has been conducted in high-income countries, where resources for health care delivery are generally more widely available than in low- and middle-income countries (LMICs). Without evidence to support the effectiveness of specific components of Collaborative Care models, policy makers in LMICs risk investing costly (and limited) resources in ineffective approaches. Key knowledge gaps exist regarding the effectiveness of Collaborative Care in LMICs and which specific model components are feasible and effective in LMIC settings. We conducted a rapid review literature search to address these concerns and identified 25 peer-reviewed published studies that evaluated the effectiveness of 20 models of Collaborative Care for depression, anxiety, schizophrenia, alcohol use disorder, and epilepsy in nine different LMICs across four World Health Organization regions. Successful models shared key structural and process-of-care elements, and there was clear evidence of the importance of tailoring the model to the local context. The review extends the literature on Collaborative Care, supports its adaptability for a broad range of disorders and its dissemination to diverse settings in LMICs, and demonstrates that more work is needed to identify strategies that will support successful and sustained implementation in LMIC clinical settings.

## Introduction

Mental disorders contribute significantly to morbidity, mortality and diminished quality of life throughout the world. A 2015 meta-analysis revealed that people with mental disorders have a mortality rate that is 2.22 times higher than the general population or people without mental disorders, with a decade of years of potential life lost (Walker et al., 2015). In 2019, depression was the second leading cause of disability worldwide, and overall, mental disorders resulted in nearly one in five years of healthy life lost due to disability (Lancet, 2020). Effective treatments exist for mental disorders, but the majority of those in need do not receive effective care (Patel et al., 2018; Thornicroft et al., 2019). In low- and middle-income countries (LMICs), it has been estimated that 79–93% of people with depression and 85–95% of people with anxiety do not have access to treatment (Chisholm et al., 2016; Evans-Lacko et al., 2018). Low availability of human resources to deliver mental health services (Kakuma et al., 2011), stigma toward mental disorders (Henderson et al., 2014), and poor implementation of mental health programs at scale contribute to this large unmet need for mental health care (Eaton et al., 2011). Globally, existing mental health services have limited capacity to address the burden of mental disorders, and the majority of mental health care is provided at the primary care level (Collins et al., 2011).

Building capacity for mental health treatment within primary care and other medical settings where people already seek care is an efficient strategy for increasing access to effective mental health treatment (Thornicroft et al., 2019). Individuals living with chronic medical conditions (such as Human Immunodeficiency Virus [HIV] and noncommunicable diseases such as diabetes and hypertension) have significantly higher rates of common mental disorders (Moussavi et al., 2007), and regular care for these conditions provides medical practitioners opportunities to identify and engage people with comorbid medical and mental disorders in care (Thornicroft et al., 2019). Integrating care for mental disorders into primary or secondary medical care can reduce the fragmentation and complexity of care, creating an opportunity for a more person-centered healthcare experience (Huang et al., 2014). Integrated care can facilitate screening for mental disorders, and increase the likelihood that people will connect to the care they need. Because integrated care models have the potential to improve quality of life, self-care, adherence to medical and mental health treatments, and both mental and physical disease outcomes (Coates et al., 2020), the World Health Organization (WHO) promotes the integration of mental health services into primary health care as a feasible strategy to address the treatment gap (Collins et al., 2011).

There are myriad approaches to integrating mental health care and primary medical care. Without evidence to support the effectiveness of specific models or approaches, policy makers risk investing costly (and limited) resources in ineffective approaches. A 2020 scoping review identified 37 models of integrated physical and mental health care published in medical literature. These models shared several key characteristics, including colocated care delivered by a multi-disciplinary team, a joint treatment plan with structured communication, and care coordination (Coates et al., 2020). Among integrated care models, the largest body of research evidence supports Collaborative Care, a complex multi-component model which applies the principles of the chronic care model (Wagner et al., 1996) to integrate evidence-based mental health treatment into outpatient medical settings. In Collaborative Care, primary medical physicians work with a care manager and a consulting psychiatrist to proactively identify, treat, and monitor people with mental disorders (Katon, 2012). Key elements include population-based patient identification; continual symptom monitoring using an electronic registry; measurement-based care to track treatment response and a stepped-care approach to systematically adjust treatment for patients who are not improving or meeting measurement-based targets (Katon et al 2012). Collaborative Care models are distinguished from other integrated care models by these core components of population-based care, measurement-based care, and delivery of evidence-based mental health services (McGinty and Daumit, 2020; Yonek et al., 2020).

The Collaborative Care model was initially designed to improve depression outcomes in primary care (Unützer et al., 2002), but over the past 20 years, it has been adapted and implemented in a wide range of mental disorders (e.g., PTSD and bipolar disorder) (Zatzick et al., 2004; Fortney et al., 2021), populations (e.g., primary care patients with diabetes) (Katon et al., 2010) and settings (e.g., maternal and child health clinics [Katon et al., 2014; Grote et al., 2015] and HIV clinics) (Pyne et al., 2011). More than 80 randomized controlled trials demonstrate Collaborative Care's effectiveness for a range of mental disorders for diverse populations and settings (Archer et al., 2012). Efforts to scale the Collaborative Care model in primary care settings, however, have not yet translated into widespread uptake or significant population health gains (McGinty and Daumit, 2020). Implementation of Collaborative Care involves substantial practice change in the medical setting. Collaborative Care models introduce both structural elements (data tracking tools and new staff, including a care manager and a psychiatric consultant) and process-of-care elements (measurement-based care (Lewis et al., 2019) and systematic caseload review (Bauer et al., 2019) to the clinical setting (McGinty and Daumit, 2020). Research suggests that tailoring evidence-based interventions to fit the clinical context is associated with increased likelihood of implementation success (Baumann et al., 2017). Across research and implementation studies in high-income countries (HICs), the Collaborative Care model core principles have been operationalized as a wide range of model components.

Less is known about whether the core components of the Collaborative Care model are feasible or effective in LMICs (Cubillos et al., 2021). Increased understanding of successful Collaborative Care models – including specific model components – is vital for policy makers and healthcare systems which seek to implement Collaborative Care to increase access to mental health care and improve outcomes in their populations (Overbeck et al., 2016; Acharya et al., 2017). A 2021 systematic review of integrated care models in LMICs identified six experimental or nonexperimental studies published between 1990 and 2017 that evaluated the effectiveness or cost-effectiveness of Collaborative Care models for depression and/or unhealthy alcohol use in LMICs (Cubillos et al., 2021). Building on these findings, we conducted a rapid review (Garritty et al., 2021) to extend the search to several mhGAP priority mental disorders and to identify more recent studies that evaluate Collaborative Care in primary or secondary outpatient medical settings in LMICs. The primary aim of this review is to describe effective Collaborative Care models that have been evaluated in LMICs and the "successful ingredients" of these models to help inform implementation by health care systems and policy makers and identify areas for future research.

## Methods

We conducted a rapid review following guidance from the Cochrane Rapid Reviews Methods Group on conducting rapid reviews (Garritty et al., 2021), and used the Preferred Reporting Items for Systematic Reviews and Meta-Analysis (PRISMA) to guide our review (Page et al., 2021). The protocol was registered on Open Science Framework (registration DOI 10.17605/OSF.IO/FJ79U).

### Electronic search strategy and sources

Our literature search strategy was informed by knowledge of the literature, discussion with knowledge experts in the field, and detailed review of published search strategies from similar literature reviews (Yonek et al., 2020; Cubillos et al., 2021). In consultation with two authors (L.C. and J.W.), a university health sciences librarian (T.J.) iteratively developed the search string and strategy, which were reviewed according to guidelines from Peer Review of Electronic Search Strategies (PRESS) (McGowan et al., 2016). A full description of the search strategy can be found in Supplementary Material. Key search terms were developed using the following sources: (1) LMIC (Cochrane Effective Practice and Organization of Care LMIC filter) and (2) integrated care (International Foundation of Integrated Care) (Lewis et al., 2018).

Our electronic search was conducted on May 23, 2022, in five databases, selected due to their focus on general medical, psychiatric and global literature: (1) PubMed, (2) Embase, (3) Global Index Medicus, (4) PsycInfo and (5) Cochrane Central. Our search was performed without language restrictions.

### Eligibility criteria

We searched for experimental and nonexperimental studies that examined the effectiveness of a Collaborative Care model on the management of any mental disorder in primary or secondary healthcare in LMICs. Included mental disorders were priority adult mental disorders in the WHO Mental Health Gap Action Programme (mhGAP) intervention guidelines (World Health Organization, 2016), which included depression, psychosis, substance use disorders, other (including anxiety and PTSD), and epilepsy (in most LMICs, epilepsy is considered a psychiatric condition and is treated by mental health specialists) (Jordans et al., 2019). Articles eligible for inclusion were required to meet the following criteria: (1) studies included patients aged ≥18 years, of any gender and with a diagnosis of mental disorder of any severity; (2) studies performed with a population living in LMICs as per the World Bank country income classification (The World Bank, 2022, *List of Low-and Middle-Income Countries*) during the year the study started; (3) studies included patients who received mental health services in an outpatient medical setting (primary or secondary health care); (4) experimental, quasi-experimental and nonexperimental study designs that reported clinical outcomes and involved a comparison group and (5) studies included Collaborative Care models, we defined to be consistent with the typology of the 2021 systematic review of integrated care models in LMICs (Cubillos et al., 2021): a multi-component, highly coordinated, team-based approach to providing mental health care with systematic integration into outpatient medical settings, with an interdisciplinary team comprised of at least a primary medical provider and a mental health care manager collaborating to systematically track patient progress and deliver evidence-based care, including pharmacotherapy, care coordination and/or brief behavioral interventions. We excluded studies that did not report clinical effectiveness outcomes, and cohort studies that reported outcomes but did not have a comparison group. We also excluded presentations, abstracts, corrections and nonpeer-reviewed papers. There were no exclusions based on language.

### Article review and selection

Article abstracts were uploaded and reviewed using Covidence Review Software. Figure 1 describes the PRISMA systematic process for article selection. Upon uploading the initial search results, Covidence automatically screened for and removed duplicate articles. Following Cochrane recommendations for rapid reviews (Garritty et al., 2021), each title and abstract was reviewed by one of the authors (J.W., S.O., A.B., B.F. and L.C.) and all excluded abstracts were reviewed again by a second author (J.W. or L.C.). Duplicates missed by Covidence's automatic process were manually marked as duplicate at this stage. Articles that met eligibility criteria and those that were inconclusive from the abstract review were included for full-text review. Protocol papers and review papers that were identified in the search were compiled and reviewed by two authors (J.W. and L.C.) to identify additional studies to include. Four authors (J.W., L.C., B.F. and S.O.) conducted full-text reviews of the studies deemed eligible based on title and abstract review. As in the title and abstract screening, articles that were excluded at this stage were reviewed by a second author and conflicts were resolved through iterative communication or discussion with the senior author (L.C.).

### Data extraction

Data were extracted by the two authors (J.W. and L.C.) utilizing a custom data extraction form in Covidence. Extracted data included: (1) study characteristics, including publication year, location (country), study design, number of participants, target mental disorder(s), and comparison treatment; (2) target population and participant characteristics; (3) intervention characteristics and model components, intervention duration; (4) primary and secondary outcomes and (5) key findings. For studies with more than one associated article, the primary article was cited as the main reference, although data were extracted from all available articles. The authors' extraction forms were compared for consistency and any differences were resolved by discussion. Components of the Collaborative Care model in each study were tracked using an intervention component checklist in the Covidence data extraction form. Model components in the intervention checklist were informed by the key components of Collaborative Care, developed by Kroenke and Unutzer (2017): mental health screening, psychiatric consultant, care manager, pharmacotherapy, measurement-based care, treatment to target, registry, brief behavioral intervention, care coordination, psychoeducation, systematic case review, systematic team communication, referral process for specialty care or stepped care. We considered a model component to be present if it was specifically mentioned, regardless of the level of detail reported.

### Assessment and synthesis

Two authors (S.O. and L.C.) independently assessed the quality of studies, using Cochrane Risk of Bias Tool version 2.0 (Sterne et al.,

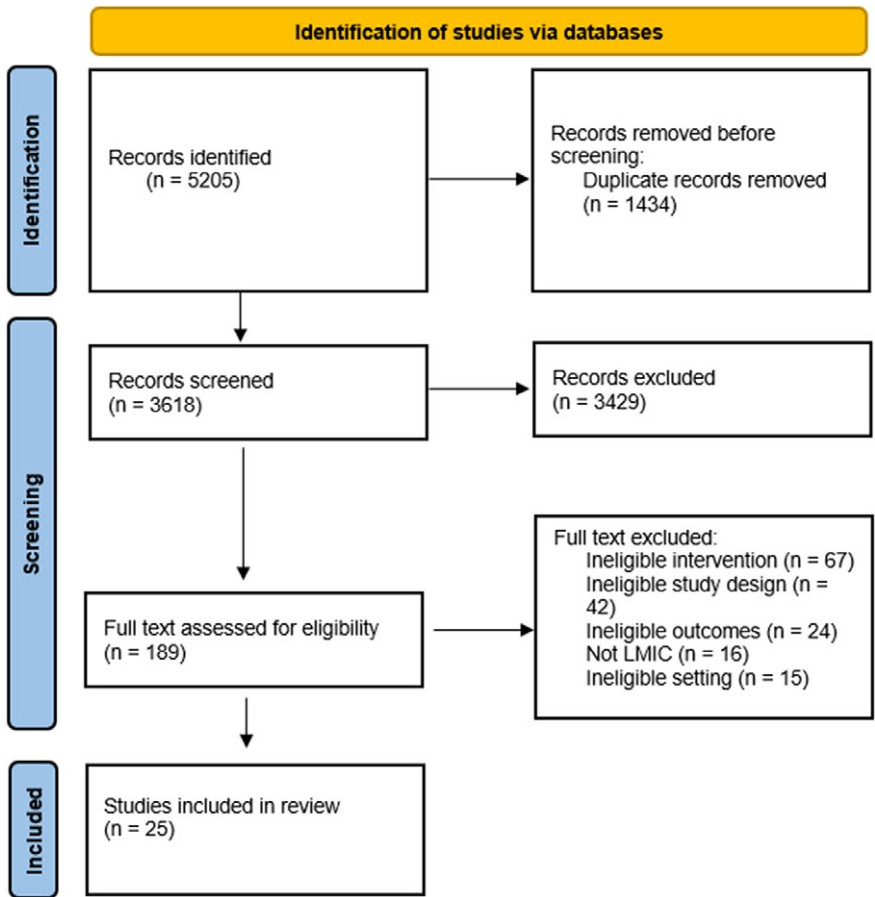

**Figure 1.** PRISMA flow diagram of studies screened and included in data extraction.

2019) for randomized controlled trials and the ROBINS-I (Risk Of Bias In Nonrandomized Studies – of Interventions) (Sterne et al., 2016) for nonrandomized studies. Discrepancies were reconciled through discussion. Descriptive statistics were used to summarize study characteristics. To synthesize the evidence supporting model components, we identified the key components within effective multicomponent interventions (Table 2) and aggregated this information across interventions to examine how frequently the components were applied (Table 3; Chorpita et al., 2007; Yonek et al., 2020).

## Results

Our search yielded 5,205 articles, from which 1,434 duplicates were removed, leaving 3,618 titles and abstracts to be reviewed. Of these, 189 were included for a full-text review. After full-text review, 25 studies met inclusion criteria, of which 20 were RCTs and 5 were cohort studies with comparison groups. We did not include for data extraction two pilot studies with subsequent RCTs that met our inclusion criteria (Oladeji et al., 2015; Adewuya et al., 2019a).

### *Overview of included studies*

Our search identified 25 studies (randomized controlled trials or nonrandomized studies with comparison groups), which described 20 Collaborative Care models. The characteristics of

these studies are summarized in Table 1. Twenty studies (Patel et al., 2010, 2011; Pradeep et al., 2014; Chen et al., 2015; Wagner et al., 2016; Indu et al., 2018; Adewuya et al., 2019b,c; Gureje et al., 2019a,b; Noorbala et al., 2019; Xu et al., 2019; Ali et al., 2020; Stockton et al., 2020; Petersen et al., 2021; Pillai et al., 2021; Asher et al., 2022; Hanlon et al., 2022; Kemp et al., 2022; Srinivasan et al., 2022) were from RCTs (primary or secondary analyses) that evaluated 16 unique models; five cohort studies (Jordans et al., 2017, 2020; Petersen et al., 2019; Shidhaye et al., 2019; Aldridge et al., 2020) evaluated an additional four Collaborative Care models. Publication dates ranged from 2010 to 2022, and sample sizes ranged from 60 patients to 2,796 patients. Nine of these models were tested in the African Region (Wagner et al., 2016; Adewuya et al., 2019c; Gureje et al., 2019a, b; Petersen et al., 2019, 2021; Stockton et al., 2020; Asher et al., 2022; Hanlon et al., 2022), eight in the South-East Asian Region (Patel et al., 2010; Pradeep et al., 2014; Jordans et al., 2017, 2020; Indu et al., 2018; Shidhaye et al., 2019; Ali et al., 2020; Srinivasan et al., 2022), two in the Western Pacific Region (Chen et al., 2015; Xu et al., 2019), and one in the Eastern Mediterranean Region (Noorbala et al., 2019). No studies were conducted in LMICs in the European Region or the Region of the Americas. (Asher et al 2022; Hanlon et al 2022; Stockton et al 2020; Wagner et al 2016).

Among the models with RCT evidence, 10 demonstrated improvement in the primary outcome (Patel et al., 2010; Chen et al., 2015; Indu et al., 2018; Adewuya et al., 2019c; Noorbala et al., 2019; Xu et al., 2019; Ali et al., 2020; Asher et al., 2022; Hanlon et al., 2022; Srinivasan et al., 2022); 5 did not (Pradeep et al.,

**Table 1.** Summary of characteristics of included randomized controlled trials and cohort studies

| | | | | | | | | | | |
|---|---|---|---|---|---|---|---|---|---|---|
| **Randomized controlled trials** | | | | | | | | | | |
| References | Country | Study years | N | Study subjects | Study disorder | Study setting | Comparison/control group | Primary outcome | Secondary analyses or study[a] | Significant effect (1° outcome) |
| Adewuya et al., 2019b[b], 2019c | Nigeria | 2014–2016 | 907 | Adults | Depression | Primary care centers | Enhanced usual care: Psychoeducation about depression, with 1 day training for staff and referral to mental health specialists as needed | Recovery (PHQ-9 score < 6) at 12 months | Pilot of text messaging intervention to increase engagement and adherence | Yes |
| Ali et al., 2020; Kemp et al., 2022[c] | India | 2015–2018 | 404 | Adults with poorly controlled diabetes | Depression | Diabetes specialty clinics | Usual care | Composite outcome of depression response (>50% improvement in SCL-20) and improvement in at least 1 cardiometabolic indicator (HbA1c, SBP or LDL cholesterol) at 24 months | Clinically significant reduction in anxiety symptoms (GAD-7) at 6 and 12 months | Yes |
| Asher et al., 2022 | Ethiopia | 2015–2017 | 149 | Adults | Schizophrenia | Primary care centers | Facility-based care: a stepped care model with mhGAP-guided medication and psychoeducation, supervision by psychiatric nurse, referral to specialty care as needed | Disability (WHODAS II) at 12 months | | Yes |
| Chen et al., 2015 | China | 2011–2013 | 326 | Adults over 60 years old | Geriatric depression | Primary care centers | Enhanced usual care: Physicians provided with guidelines on depression treatment and screening results (PHQ9, dx of depression) | Depression response (HAMD-17) at 3, 6 and 12 months | | Yes |
| Gureje et al., 2019a | Nigeria | 2013–2015 | 686 | Antenatal women | Perinatal depression | Primary maternal care centers | Enhanced usual care (low-intensity treatment): mhGAP-guided depression treatment, with psychosocial intervention and psychoeducation but no stepped care protocol | Depression remission at 6 months postpartum, defined as an EPDS <6. | | No |

**Table 1.** (*Continued*)

| | | | | | | | | | | |
|---|---|---|---|---|---|---|---|---|---|---|
| **Randomized controlled trials** | | | | | | | | | | |
| References | Country | Study years | N | Study subjects | Study disorder | Study setting | Comparison/control group | Primary outcome | Secondary analyses or study[a] | Significant effect (1° outcome) |
| Gureje et al., 2019b | Nigeria | 2013–2015 | 1,178 | Adults | Depression | Primary care centers | Enhanced usual care: mhGAP-guided depression treatment with 2-day training and results of PHQ9 screen | Depression remission (PHQ-9 score of <6) at 12 months | | No |
| Hanlon et al., 2022 | Ethiopia | 2015–2016 | 324 | Adults over 25 yo | Schizophrenia, primary psychotic disorders, schizoaffective disorder, bipolar disorder or severe MDD | Primary care centers | Outpatient psychiatric nurse care augmented with 2 days mhGAP training and 1 day training in management of SMD in pregnant or breastfeeding women. Community-based lay project workers also supported engagement | Clinical symptom severity (BPRS-E) at 12 months | | Yes[d] |
| Indu et al., 2018 | India | 2012–2014 | 60 | Adult women | Moderate to severe depression | Primary care centers | Usual care: Referral to hospital or private practice mental health services | Depression response (HAMD) at 8 weeks | | Yes |
| Noorbala et al., 2019 | Iran | 2015–2018 | 202 | Adult pregnant women | Perinatal depression | Primary care centers | Usual care: routine pregnancy treatment | Improvement of physical and anxiety symptoms (GHQ-28) at 35–37 weeks of pregnancy, 6 weeks postpartum and 6 months postpartum | | Yes |
| Patel et al., 2010, 2011; Pillai et al., 2021[c] | India | 2007–2010 | 2,796 | Adults | Depressive and anxiety disorders | Primary care centers and private GP practices | Usual care: Screening results were provided to physicians, who were given a treatment manual and initiated treatments of their choice | Recovery from depression and anxiety (ICD-10 diagnosis via CIS-R) at 6 months | Secondary analysis included suicide attempts/plans (CIS-R); days out of work, psychological morbidity and overall disability (WHODAS II) | Yes |

(*Continued*)

**Table 1.** (*Continued*)

| | | | | | | | | | Secondary analyses or study[a] | Significant effect (1° outcome) |
|---|---|---|---|---|---|---|---|---|---|---|
| Randomized controlled trials | | | | | | | | | | |
| References | Country | Study years | N | Study subjects | Study disorder | Study setting | Comparison/control group | Primary outcome | | |
| | | | | | | | | | Secondary analysis of self-reported antidepressant adherence | |
| Petersen et al., 2021 | South Africa | 2015–2016 | 1,043 | Adults with HTN | Depression | Primary care centers | Usual care, including referral to primary health center doctors and/or mental health specialists | Depression response (50% reduction in PHQ-9) at 6 months | | No |
| Pradeep et al., 2014 | India | 2006–2009 | 260 | Adult women | Depression | Primary care centers | Usual care: Patients diagnosed with depression were encouraged to seek help from the physician at PHC with no additional input from the community health worker | Number who sought and completed treatment; changes in severity of depression (HDRS), and WHO-QOL (Brev) scale at 6 months | | No |
| Srinivasan et al., 2022 | India | 2015–2018 | 2,486 | Adults with HTN, diabetes, and/or ischemic heart disease, over 30 yo | MDD, dysthymia, GAD, and/or panic disorder | Primary care centers | Enhanced usual care: primary health clinicians received basic training in the treatment of depression; referral to specialty treatment for patients with moderate or greater depression or high suicide risk | Severity of depression (PHQ-9) at 3, 6 and 12 months | | Yes |
| Stockton et al., 2020 | Malawi | 2017–2018 | 501 | Nonpregnant adults living with HIV | Depression | HIV clinics | Control phase: Providers screened for depression using PHQ9, with suicide risk assessment protocol (SRAP) and hospital referral when needed | Retention in HIV care over 6 months; viral suppression at 6 months | Depression remission (PHQ-9 < 5 at 5 months) | No |

(*Continued*)

**Table 1.** (*Continued*)

| Randomized controlled trials | | | | | | | | | |
|---|---|---|---|---|---|---|---|---|---|
| References | Country | Study years | N | Study subjects | Study disorder | Study setting | Comparison/control group | Primary outcome | Secondary analyses or study[a] | Significant effect (1° outcome) |
| Wagner et al., 2016 | Uganda | 2013–2014 | 1,252 | Adults living with HIV | Depression | HIV clinics | Clinical acumen model: The same screening, monitoring and supervision as intervention, but PCPs used clinical judgment for further evaluation and treatment of depression | Depression response (PHQ-9) at 12 months | | No |
| Xu et al., 2019 | China | 2015–2016 | 278 | Adults | Schizophrenia | Community and primary care centers | Free national antipsychotic medication program | Medication adherence (proportion of dosages taken) at 6 months | Psychosis symptoms (CGI for schizophrenia) | Yes |

| Cohort studies with comparison groups | | | | | | | | | |
|---|---|---|---|---|---|---|---|---|---|
| References | Country (urban vs. rural) | Study years | N | Study subjects | Study disorder | Study setting | Comparison/control group | Primary outcomes | Secondary analyses or study[a] | Significant effect (1° outcome) |
| Jordans et al., 2017 | Nepal | 2013–2015 | 204 | Adults | Epilepsy and psychotic disorders | Primary care centers | Enhanced usual care: a comparison group was offered a more basic set of services consisting only psychotropic medicines by mhGAP-trained health workers | Psychosis symptoms (PANSS); Seizures (Epilepsy-9); Disability (WHODAS) all at 12 months. | | Yes |
| Aldridge et al., 2020; Jordans et al., 2020 | Nepal | 2014–2016 | 438 | Adults | Depression, and AUD | Primary care centers | Usual care: standard primary care with no mental health treatment provided | Change in depression (PHQ9) and AUD symptoms (AUDIT-10), and disability (WHODAS) at 3 and 12 months | Aldridge 2020: Decrease in suicidal ideation for depression and AUD cohorts | Yes |
| Petersen et al., 2019 | South Africa | 2014–2016 | 373 | Adults with chronic illness (HIV, TB, HTN) | Depression and AUD | Primary care centers | Usual care | Provider identification of depressive and AUD symptoms at 12 months; response and remission of depressive symptoms at 3 and 12 months (PHQ9) | | Yes |

(*Continued*)

**Table 1.** *(Continued)*

**Cohort studies with comparison groups**

| References | Country (urban vs. rural) | Study years | N | Study subjects | Study disorder | Study setting | Comparison/control group | Primary outcomes | Secondary analyses or study[a] | Significant effect (1° outcome) |
|---|---|---|---|---|---|---|---|---|---|---|
| Shidhaye et al., 2019 | India | 2014–2016 | 748 (in cohort study) | Adults, including perinatal women | Depression (including maternal depression), psychosis and AUD | Primary care and maternal care centers | Usual care for general health complaints from the medical officer and no mental health interventions | At 26 months, change in contact coverage (i.e, difference in the proportion of individuals with depression or AUD (PHQ-9 > 10 or AUDIT score > 8) who sought treatment | Response, early remission and recovery (PHQ9 and AUDIT) | No for primary outcome, yes for clinical outcomes |

Abbreviations: AUD, alcohol use disorder; AUDIT, Alcohol Use Disorders Identification Test; ART, antiretroviral therapy; BPRS-E, Brief Psychiatric Rating Scale-Expanded; CGI, Clinical Global Impressions Scale; CIS-R, Clinical Interview Scale-Revised; Epilepsy-9, 9-item instrument re: the number epileptic seizures in the previous 3 months; GAD-7, Generalized Anxiety Disorder 7 item scale; GHQ-28, General Health Questionnaire 28 items; GRIMS, Golombok Rust Inventory of Marital State; HAMD, Hamilton Depression Rating Scale; HIV, human immunodeficiency virus; HTN, hypertension; HDRS, Hamilton Depression Rating Scale; PANSS, Positive and Negative Syndrome Rating Scale; PHQ9, Patient Health Questionnaire 9 item; SCL-20, Symptom Checklist-20 item; TB, tuberculosis; WHODAS, World Health Organization Disability Assessment Schedule.
[a]Follow-up duration the same as primary outcome unless otherwise noted.
[b]Primary outcome of comparison between Collaborative Care intervention and Collaborative Care intervention with mobile telephony support.
[c]Secondary analyses.
[d]Noninferiority trial.

2014; Wagner et al., 2016; Gureje et al., 2019b; Stockton et al., 2020; Petersen et al., 2021). One study compared a high-intensity intervention to a low-intensity intervention; both interventions improved clinical outcomes, but there was no additional benefit to the high-intensity intervention (Gureje et al., 2019a). All four cohort studies demonstrated improved clinical outcomes among patients who received Collaborative Care when compared to the comparison group.

The majority of models were tested in general primary care settings, but two were tested in outpatient HIV clinics/treatment centers (Wagner et al., 2016; Stockton et al., 2020), one in diabetes specialty clinics (Ali et al., 2020) and two in maternal health clinics (Gureje et al., 2019a; Shidhaye et al., 2019). Seventeen of the 20 models targeted depression, with three that focused on maternal/perinatal depression (Gureje et al., 2019a; Noorbala et al., 2019; Shidhaye et al., 2019), and one on geriatric depression (Chen et al., 2015). Five models targeted either schizophrenia or psychosis or more serious mental disorders (Jordans et al., 2017; Shidhaye et al., 2019; Xu et al., 2019; Asher et al., 2022; Hanlon et al., 2022), three targeted alcohol use disorder (Petersen et al., 2019; Shidhaye et al., 2019; Jordans et al., 2020), and one targeted epilepsy (Jordans et al., 2017). Three models either targeted anxiety disorders or were shown to have positive impact on anxiety (Patel et al., 2010; Kemp et al., 2022; Srinivasan et al., 2022). No studies addressed PTSD or substance use disorders other than alcohol. Primary outcomes for the studies were validated clinical rating scales in 17 of the studies, disability for 3 studies, and treatment or medication adherence for 4 studies. The most common measures were Patient Health Questionnaire-9 (PHQ-9) (Kroenke et al., 2001) for depression, WHO Disability Assessment Scale II (WHODAS II) (Chwastiak and Von Korff, 2003) for disability, Positive and Negative Symptom Scale (PANSS) (Kay et al., 1987) for schizophrenia, and AUDIT-C (Bradley et al., 2007) for alcohol use disorder.

Fourteen of the studies reported the primary results of randomized controlled trials. Nine of these were positive studies (Patel et al., 2010; Chen et al., 2015; Indu et al., 2018; Noorbala et al., 2019; Xu et al., 2019; Ali et al., 2020; Asher et al., 2022; Hanlon et al., 2022; Srinivasan et al., 2022), and eight were assessed to have low risk of bias. One positive trial was stopped early (at 12 months rather than the planned 24) and missing data were not imputed, introducing the potential for bias (Chen et al., 2015). Among the five RCTs that did not have a significant impact on the primary outcome (Pradeep et al., 2014; Wagner et al., 2016; Gureje et al., 2019b; Stockton et al., 2020; Petersen et al., 2021), there was concern of risk of bias in favor of the comparison condition in one of the studies. This pragmatic study may have been impacted by the cointervention of concentrating referral specialist mental health services in the control clinics to improve service coverage in the district (Petersen et al., 2021). The Stockton trial was assessed to have low risk of bias (and included intent-to-treat analyses), but authors noted that few participants received an adequate dose of either pharmacotherapy or the psychological intervention (Stockton et al., 2020). Four of the five included nonrandomized cohort studies recruited comparison samples from patients who had screened positive as part of the intervention workflow, but whose diagnosis was not detected by the medical provider (Petersen et al., 2019; Shidhaye et al., 2019; Aldridge et al., 2020; Jordans et al., 2020). This may have introduced of bias in favor of the intervention as the screening interview may have heightened patient awareness of their symptoms.

### Components of Collaborative Care models

Table 2 provides an overview of the models and components for the included studies, and Table 3 summarizes the frequencies of each of these components across models from these studies. *Team-based care* is a required component of Collaborative Care, but the models varied with respect to the composition of the team. In nine models, the care manager role was filled by a lay health worker or community health worker (Patel et al., 2010; Pradeep et al., 2014; Jordans et al., 2017, 2020; Gureje et al., 2019b; Shidhaye et al., 2019; Xu et al., 2019; Stockton et al., 2020; Hanlon et al., 2022); nurses filled the role in four other models (Chen et al., 2015; Wagner et al., 2016; Indu et al., 2018; Srinivasan et al., 2022). Four models utilized other clinical staff, including midwives (Adewuya et al., 2019c; Noorbala et al., 2019), other maternal health care providers (Gureje et al., 2019a), or allied health professionals working in diabetes clinics (Ali et al., 2020). The roles and tasks of care managers were split across multiple team members in four of the models (Adewuya et al., 2019c; Petersen et al., 2019, 2021; Asher et al., 2022). Thirteen studies included a consulting psychiatrist who provided regular consultation to either the care manager, the primary care physician or both; frequency of consultation ranged from every week to every month (Adewuya et al 2019c; Ali et al 2020; Chen et al 2015; Gureje et al 2019a; Gureje et al 2019b; Hanlon et al 2022; Jordans et al 2017; Noorbala et al 2019; Patel et al 2010; Shidaye et al 2019; Srinivasan et al 2022; Wagner et al 2016; Xu et al 2019). Two models included a pharmacist or pharmacy technician on the Collaborative Care team (Adewuya et al., 2019c; Srinivasan et al., 2022).

All models included *evidence-based treatments* for the target mental disorder. In all cases, this included pharmacotherapy, which was supported by mhGAP or national treatment guidelines. One study included electronic decision support within the medical record to support physician prescribing (Ali et al., 2020). Psychoeducation or brief psychological interventions were included in all studies, either in individual or group format; and in all models, these were delivered by the care manager. Behavioral activation and Problem-Solving Therapy were the most common brief psychological interventions for studies of depression. All models described training and supervision protocols, highlighting the critical need for staff and resources for these activities.

*Population management* components commonly included universal/ routine screening for the target mental disorder and specific strategies for outreach to patients who were not engaged in care. Fourteen of the 20 models provided specifics about a stepped approach to care: measurement at specified time intervals with treatment intensification (either increase in number of sessions of psychological intervention, combination of pharmacotherapy and psychological intervention; or referral to mental health specialist) (Table 2). The greatest variation across studies was with respect to how measurement was incorporated into the clinical workflow. Eleven studies explicitly described measurement-based care, that is, regularly scheduled follow-up by the care manager and regular tracking of a validated clinical outcome measure (Adewuya et al 2019c; Ali et al 2020; Chen et al 2015; Gureje et al 2019a; Gureje et al 2019b; Noorbala et al 2019; Patel et al 2010; Shidaye et al 2019; Stockton et al 2020; Wagner et al 2016; Xu et al 2019). Treatment-to-target, though, was present in fewer than half of the studies (Adewuya et al 2019c; Ali et al 2020; Chen et al 2015; Gureje et al 2019a; Gureje et al 2019b; Noorbala et al 2019; Patel et al 2010; Shidaye et al 2019). Only two studies described the use of a registry to support the clinical workflow (Wagner et al., 2016; Ali et al., 2020). Few models included mobile or digital support systems to support patient communication and engagement (Adewuya et al 2019b; Gureje et al 2019a; Xu et al 2019) (Table 3).

### Discussion

This rapid review supports the effectiveness of Collaborative Care models to treat a wide range of mental disorders in diverse outpatient medical settings in LMICs. We identified 25 randomized controlled trials or cohort studies with comparison groups which evaluated the effectiveness of 20 Collaborative Care models to treat common mental disorders, schizophrenia, alcohol use disorder, or epilepsy in nine different LMICs. Fourteen of the 20 Collaborative Care models had statistically significant improved clinical outcomes compared to usual primary care. Clinical outcomes were primarily validated rating scales of symptom severity or disability. More recent studies, specifically models that provided treatment for schizophrenia, highlighted the critical need to address social determinants of health, monitor functional outcomes, and link clinic-based care with community- or family-based services. Effectiveness data from randomized controlled trials, however, was limited to studies of common mental disorders (depression and anxiety) or schizophrenia. No RCT of Collaborative Care interventions for substance use disorders, epilepsy or post-traumatic stress disorder were identified.

Despite differences in staffing and resources across the clinical settings in these studies, each of these models operationalized the same core principles of effective Collaborative Care that are described in studies in HIC (Sighinolfi et al., 2014; Muntingh et al., 2016; Dham et al., 2017; Yonek et al., 2020). As in HIC studies, effective models shared several structural and process-of-care elements. Structural elements included a multi-disciplinary care team and standardized protocols for the delivery of evidence-based (pharmacologic and/or brief psychological intervention). Shared process-of-care elements included proactive and systematic identification of mental disorders, team-based care with structured communication, and longitudinal measurement of patient response to treatment and a stepped-care approach to intensify treatment when measurements show that a patient is not improving as expected. Some core components of Collaborative Care models implemented in HIC were less frequently described in these LMIC studies. Specifically, relatively few models described rigorous measurement-based care and the systematic use of a registry to support the clinical workflow.

There was, however, substantial heterogeneity across models and their components. This is consistent with experience in implementation of other evidence-based interventions that there is a need to tailor interventions for specific target populations and clinical contexts (Wiltsey Stirman et al., 2012). For example, a wide range of disciplines performed the role of the care manager, and the frequency and modes of team communication ranged widely. Almost all studies included psychiatric expertise on the multi-disciplinary team, and a regular structured meeting to review the caseload of patients was typical of the effective models. In addition, linkage to community resources is a core component of Collaborative Care models, but specific tasks and workflows were dependent on the specific context. Several of the studies highlighted that tailoring the model to both culture and clinical context was critical for its effectiveness (Kemp et al., 2022). Several studies described how Collaborative Care can promote culturally appropriate care, and that collaborations with other community sectors can address social and economic determinants of mental health.

**Table 2.** Summary of published components of Collaborative Care models

| Intervention | Target conditions | Team | Evidence-based treatments | Measurement | Caseload review | Stepped care | Population management |
|---|---|---|---|---|---|---|---|
| *Mental Health in Primary Care Project (MeHPriC-P)* Adewuya et al., 2019a, 2019b, and 2019c (Nigeria) | Depression | • Primary health care center medical doctor who prescribed medication<br>• Nurses/midwives trained to deliver PST-PC<br>• Community Health Officers<br>• Community Health Extension Workers<br>• Pharmacy technicians.<br>• Mental health specialist team | • Psychoeducation<br>• Antidepressant medication (for severe depression, PHQ-9 > 14, or refusing PST-PC)<br>• Problem-Solving Therapy: 10 sessions in 14 weeks, delivered by trained staff nurse | • Patients were reassessed with PHQ-9 after the 3rd and 6th PST sessions<br>• Antidepressant response was reassessed after 6 weeks | • Mental health team provided clinical support and supervision via mobile telephone and monthly site visits | • Patients without significant improvement (PHQ-9) were offered PST-PC with anti-depressant<br>• Patients with psychosis or suicidality were referred to mental health team | • In pilot of mobile telephony-supported model, patients received text message reminders of appointments, treatment adherence and relapse prevention |
| *Integrating Depression and Diabetes Treatment (INDEPENDENT)* Ali et al., 2020; Kemp et al., 2022 (India) | Depression and anxiety | • Care coordinator (CC) (nutritional counselors)<br>• Two consulting specialists (psychiatrist and endocrinologist)<br>• Diabetes clinic physicians prescribed all medications | • Psychoeducation<br>• Pharmacotherapy informed by national guidelines and clinical decision-making supported by decision-support software<br>• Behavioral Activation | • Regular (every 2–4 weeks) phone or in-person contact with CC<br>• PHQ-9 and review of glucose and blood pressure logs at every contact | • Care coordinator and consulting specialists met for caseload review every 2–4 weeks<br>• Treatment-to-target | • Increased risk or complexity referred to psychiatrist | • Outcomes were tracked in an electronic clinical registry<br>• Systematic outreach for those not engaged in care |
| *Rehabilitation Intervention for People with Schizophrenia in Ethiopia (RISE)*, implemented as part of the Programme for Improving Mental Health Care (PRIME) Asher et al., 2022 (Ethiopia) | Schizophrenia | • Nurses trained in psychoeducation<br>• Health officers trained in prescribing and managing anti-psychotic medication<br>• CBR workers (lay people from community) provided psychoeducation, counseling, supported adherence, monitored medication side effects, suicidality or relapse and ran family support groups<br>• Psychiatric nurse | • Psychoeducation<br>• Antipsychotic medication<br>• Problem-Solving Therapy | • Not described | • Psychiatric nurse-supervised care | • Frequency of contact was based on clinical progress<br>• Primary care staff referred patients to psychiatric nurse-led outpatient care, inpatient care or health center for suicide risk, relapse, or medication side-effects | • CBR workers mobilized resources from community for patients and supported facility-based care engagement |
| *Depression Care Management (DCM)* Chen et al., 2015 (China) | Geriatric depression | • Care manager (nurse) educated patients and families about depression, facilitated communication between patients and providers | • Psychoeducation<br>• Pharmacotherapy based on treatment algorithm<br>• Depression Care Management | • Care manager administered PHQ-9 every two weeks for 16 weeks<br>• PCPs made medication adjustments | • Monthly clinic visits from psychiatrist supported team function, education and program implementation, and provided consultation | • Referral to specialty mental health clinic | • Care manager supported adherence to treatment via telephone calls every 2 weeks |

(*Continued*)

| Intervention | Target conditions | Team | Evidence-based treatments | Measurement | Caseload review | Stepped care | Population management |
|---|---|---|---|---|---|---|---|
| | | • Primary care physician prescribed antidepressants, saw patients every 2 weeks<br>• Consulting psychiatrist | | based on PHQ-9 at each visit | on patients not improving | | |
| *EXPONATE*<br>Gureje et al., 2019a<br>(Nigeria) | Perinatal depression | • Primary maternal care provider (PMCP) delivered PST and monitoring<br>• Primary care physicians supervised PMCP and consulted psychiatric consultant when necessary<br>• Psychiatrist provided consultation as needed | • Psychoeducation<br>• Problem-solving treatment (PST) for Primary Care<br>• Pharmacotherapy<br>• Parenting skills training<br>• Structured sessions for high-intensity treatment model (HIT) only | • Screening with EPDS<br>• Progress through model was determined by scores on EPDS, time since enrollment and gestational status | • The support, supervision and specialist consultation were provided via mobile phones except when face-to-face assessment was indicated between PCP and PMCP or PCP and psychiatrist. | • For HIT only: Steps included increased frequency of PST, antidepressant, or referral to mental health specialty care | • Screening with EPDS<br>• Participants received automated mobile phone messages from PMCP with appt and PST homework reminders |
| *STEPCARE*<br>Gureje et al., 2019b<br>(Nigeria) | Depression | • Primary care providers (nurses and community health officers, or CHEWs) were trained in and delivered pharmacotherapy, psychoeducation, behavioral activation and PST<br>• General practitioner trained in mhGAP pharmacotherapy acted as primary health care coordinator and provided supervision<br>• Psychiatrist | • Psychoeducation<br>• Pharmacotherapy<br>• Behavioral activation<br>• Problem-solving therapy | • PHQ-9 screen scores to determine treatment options (PST for scores 11–14, vs. antidepressant for scores >15)<br>• PHQ-9 scores were reassessed after eight sessions | • General practitioner trained in mhGAP pharmacotherapy provided supervision/consultation for antidepressant management<br>• Psychiatrist discussed cases for those not improving after combination trial<br>• Supervision and consultations were also provided as-needed, and through mobile phones, except when a face-to-face was necessary | • Those not improving with treatment received PST plus antidepressant or had case discussed with psychiatrist | • Screening with PHQ-9 |
| *Task-shared care for severe mental disorders (TaSCS)*<br>Hanlon et al., 2022<br>(Ethiopia) | Severe mental disorders (schizophrenia, primary psychotic disorders, severe depression, schizoaffective disorder and bipolar disorder) | • Nonphysician primary care workers (eg, health officers and nurses) provided pharmacotherapy<br>• Community-based health extension workers (CHEW) trained in mh-GAP to provide psychosocial intervention and outreach | • Psychoeducation<br>• PHC workers were expected to follow one-page care plan guided by mhGAP-IG and project psychiatric nurse to provide ongoing psychiatric care<br>• Pharmacotherapy | • Not described | • Health officers, nurses and CHEWs received ongoing group supervision by project psychiatric nurse for treatment of severe mental disorders weekly to 2 weeks for the first 1.5 years, then monthly | • Referral to a psychiatric clinic if needed | • Community engagement and outreach by CHEWs for people who dropped out of care |

**Table 2.** (*Continued*)

| Intervention | Target conditions | Team | Evidence-based treatments | Measurement | Caseload review | Stepped care | Population management |
|---|---|---|---|---|---|---|---|
| *Community-based Depression Intervention Programme (ComDIP)* Indu et al., 2018 (India) | Moderate to severe depression | • Medical health officers prescribed medication, reassessed (weekly for the first two weeks, then once in two weeks), and discussed side effects, symptoms or adjusted medication as needed<br>• Care manager (Junior Public Health Nurse) delivered education, supported adherence, provided behavioral activation, monitored symptoms and coordinated care | • Depression psychoeducation<br>• Behavioral activation<br>• Pharmacotherapy | • Screening with PHQ-9 | • Not described | • Not described | • Screening with PHQ-9 |
| *The Mental Health Beyond Facilities (mhBeF) Project* Jordans et al., 2017 (Nepal) | Epilepsy and psychosis | • Primary healthcare workers prescribed and managed psychotropic medication<br>• Community counselors delivered psychosocial support through individual or family counseling and patient support groups<br>• Female community health volunteers provided stigma reduction and patient engagement | • Psychoeducation<br>• Psychosocial support and pharmacotherapy guided by mhGAP | • Screening with brief screening tool developed for study | • Psychiatrists provided supervision to primary health care workers | • Not described | • Female community health volunteers ensured follow-up care through home-based care<br>• A newly developed procedure (Community Informant Detection Tool – CIDT) was used for pro-active case detection in community |
| *Mental Health Care Plan (MHCP) as part of PRIME* Aldridge et al., 2020; Jordans et al., 2020; (Nepal) | Depression and AUD | • Female health community volunteers pro-actively identified cases and delivered home-based care<br>• Community counselors delivered HAP and CAP<br>• Health workers trained in mhGAP to diagnose, provide psychoeducation, and pharmacotherapy | • Psychoeducation<br>• Healthy Activity Program (HAP) for depression, consisting of 6–8 weekly sessions with BA<br>• Counseling for Alcohol Problems (CAP) for AUD, a manualized MI intervention of 4 weekly sessions | • Screening with PHQ-9 and AUDIT | • Not described | • Referrals to specialized care as needed | • Screening with PHQ-9 and AUDIT<br>• Female health community volunteers pro-actively identified cases and delivered home-based care |
| Noorbala et al., 2019 (Iran) | Perinatal depression | • Physicians received training in diagnosis, treatment | • Psychoeducation<br>• Supportive psychotherapy<br>• Pharmacotherapy | • GHQ-28 was used for screening, determining risk group, and was administered at | • The study psychiatrist provided feedback or evaluation of high-risk | • Patients were stratified into risk groups (low, medium and high) based on | • GHQ-28 was used for screening |

(*Continued*)

**Table 2.** (*Continued*)

| Intervention | Target conditions | Team | Evidence-based treatments | Measurement | Caseload review | Stepped care | Population management |
|---|---|---|---|---|---|---|---|
| | | • Midwives received training and delivered stress management and psychoeducation to low-risk groups<br>• General practitioners delivered supplementary stress management, supportive psychotherapy, and family conflict resolution to medium and high-risk groups, and coordinated care with midwives based on patient progress<br>• The study psychiatrist provided case consultation to general practitioners | | four different times: 6–10 weeks of pregnancy, 35–37 weeks of pregnancy, 6 weeks after delivery, and 6 months after delivery.<br>• Marital satisfaction using the GRIMS | cases to general practitioners | GHQ-28, family history and psychiatric history, and intensity of treatment adjusted based on risk group at baseline and reassessment<br>• Patients in high-risk group were referred to study psychiatrist | |
| *MANAS Collaborative Stepped Care (CSC):* Patel et al., 2010, 2011; Pillai et al., 2021 (India) | Common mental disorders (CMD) (depression and anxiety) | • LHC-screened patients provided psychoeducation, case management, psychosocial interventions and coordinated care with PCP<br>• PCP prescribed and monitored antidepressant medication<br>• A visiting psychiatrist (clinical specialist) visited the clinic monthly to provide psychiatric consultation of the caseload) who all work in close collaboration. Medical issues | • Psychoeducation<br>• Pharmacotherapy<br>• Adapted interpersonal therapy | • GHQ-28 used for screening, determining psychosocial intervention versus medication, and monitoring symptoms | • The clinical specialist visited the clinic monthly to provide psychiatric consultation of the caseload | • For patients with moderate to severe CMD (i.e., GHQ score > 7) or for those who did not respond to psycho-education alone with LHC, antidepressants and/or interpersonal therapy were offered<br>• Referral to the psychiatrist for those with high suicide risk, not responding to treatments, consultation requests by the PCP, or those with diagnostic questions, substance use issues or significant comorbid medical issues | • GHQ-28 used for screening |
| *PRogramme for Improving Mental Health CarE (PRIME)* Petersen et al., 2019 (South Africa) | Depression and AUD | • Primary health care nurses functioned as case managers, were oriented to protocols, trained to identify and provide follow-up assessment and medication management for depression, | • Psychoeducation<br>• Decision support tool guidelines (Adult Primary Care)<br>• Counseling based on CBT and PST<br>• Pharmacotherapy | • PHQ-9 and AUDIT used for screening | • Supervision of clinic-based lay counselors by district psychology team | • Referral for mental health specialist care for acute psychosis, suicide risk or alcohol dependence | • PHQ-9 and AUDIT used for screening<br>• Case detection by community health workers |

(*Continued*)

| Intervention | Target conditions | Team | Evidence-based treatments | Measurement | Caseload review | Stepped care | Population management |
|---|---|---|---|---|---|---|---|
| | | other mental health conditions<br>• Primary care doctors trained to initiate and manage antidepressants<br>• Clinic-based lay counselors trained to deliver individual and group-based counseling<br>• Community health workers involved in case detection | | | | | |
| *PRogramme for Improving Mental Health CarE (PRIME)*<br>Petersen et al., 2021 (South Africa) | Depression | • Primary health center nurses trained in communication and mental health content<br>• Primary health care physicians trained to manage antidepressants<br>• Lay counselors trained to provide evidence-based therapy<br>• Community health workers involved in case detection | • Psychoeducation<br>• Counseling based on CBT and PST<br>• Decision support tool guidelines (Adult Primary Care)<br>• Pharmacotherapy | • Screening with PHQ-9 | • Supervision of clinic-based lay counselors by district psychology team | • Referral for patients endorsing suicidal ideation or SMI were referred to specialty mental health or hospitalization | • Screening with PHQ-9<br>• Community health workers involved in case detection |
| Pradeep et al., 2014 (India) | Depression | • Primary health center doctor provided monthly visits<br>• Community Health Worker who educated patients and families, supported adherence and provided follow-up to monitor adherence, side effects and response | • Pharmacotherapy<br>• Psychoeducation | • Screening with GHQ-28<br>• Home visits monitored pill counts | • Not described | • Not described | • Screening with GHQ-28<br>• CHWs visited patients who discontinued medication and / or did not visit the PHC |
| *PRogramme for Improving Mental Health CarE (PRIME)*<br>Shidhaye et al., 2019 (India) | Depression (including maternal depression), psychosis and AUD | • Case managers provided screening, psychoeducation, care coordination with medical officers, and mental health first aid to people in community<br>• Accredited Social Health Activist (ASHAs) met with case | • Psychoeducation<br>• Healthy Activity Program (HAP) for depression, with BA<br>• Counseling for Alcohol Problems (CAP) for AUD, with MI<br>• Pharmacotherapy | • Screening with PHQ-9 and AUD<br>• Patients with PHQ-9 > 14 received both anti-depressants and counseling while those with PHQ-9 ≤ 14 or perinatal women received only counseling | • District Mental Health Programme Psychiatrist visited once a month to provide consultation for severe cases, especially those with psychosis | • Not described | • Health Management Information System to collect process data was developed<br>• Case managers maintained regular follow up with patients<br>• Case managers also visited villages and coordinated with |

**Table 2.** (*Continued*)

| Intervention | Target conditions | Team | Evidence-based treatments | Measurement | Caseload review | Stepped care | Population management |
|---|---|---|---|---|---|---|---|
| | | • managers to identify those with mental health issues<br>• Medical officers confirmed diagnoses and prescribed anti-depressants or other psychotropic medication | | | | | ASHAs to identify those in need of MHFA or further evaluation |
| *HOPE (Healthier Options through Empowerment)* Srinivasan et al., 2022 (India) | Major depressive disorder, dysthymia, generalized anxiety disorder, and/or panic disorder | • Primary care physician prescribed anti-depressant medications<br>• Care manager (nurse) tracked patient progress<br>• Community health worker facilitated lifestyle group<br>• Pharmacist provided education about medications<br>• Consulting psychiatrist and community medicine physician | • Psychoeducation<br>• Pharmacotherapy for depression and anxiety<br>• Community-based weekly Healthy Living Group | • Not described | • Consulting psychiatrists provided weekly consultations to primary care physicians for difficult cases | • High-risk for suicide was referred to district hospital<br>• Abnormal laboratory values referred to relevant specialists | • Community Health workers acted as liaison between patients and families and treatment team |
| Stockton et al., 2020 (Malawi) | Depression | • ART providers (nurses or clinicians) screened patients and prescribed antidepressants<br>• Clinic-based lay health workers or project employed counselor delivered Friendship Bench therapy | • Psychoeducation<br>• Behavioral activation<br>• Adapted PST as Friendship Bench therapy<br>• Pharmacotherapy using algorithm-based care and depression management protocol | • PHQ-9 scores used to triage patients by depressive severity<br>• Those with scores PHQ-9 5–9 were referred to Friendship-Bench therapy<br>• Those with PHQ-9 score > 10 received antidepressants<br>• Patients were reevaluated at least monthly with PHQ-9 to guide treatment titration | • Not described | • PHQ-9 scores used at ART initiation to triage patients by depressive severity | • PHQ-9 used for screening |
| *INDEPTH (INtegration of DEPression Treatment in HIVcare)* Wagner et al., 2016 (Uganda) | Depression | • Expert clients/village health team workers did initial screening<br>• Depression care nurse screened with PHQ-9 and MINI, provided psychoeducation and pharmacotherapy with antidepressant, | • Pharmacotherapy dependent on patient symptoms and psychiatric history<br>• Psychoeducation | • Initial screening with PHQ-2 and those positive were screened with PHQ-9 and MINI<br>• PHQ-9 in follow-up every 2 weeks to gauge treatment response | • Monthly ongoing, on-site supervision, caseload review and program monitoring from study psychiatrists was provided one-on-one with depression care nurses using the | • Not described | • Screening with PHQ2, PHQ-9 and MINI<br>• A Depression Treatment Registry was used to record medication, dosage and other data clinical data for each |

**Table 2.** (*Continued*)

| Intervention | Target conditions | Team | Evidence-based treatments | Measurement | Caseload review | Stepped care | Population management |
|---|---|---|---|---|---|---|---|
| | | followed up and reassessment of side effects, adherence and symptoms | | • Follow-up frequency (1–3 months) was based on response | Depression Treatment Registry | | visit, and was reviewed during supervision, follow-ups and for program monitoring |
| *LEAN (Lay health supporters, E-platform, Award, and iNtegration)* Xu et al., 2019 (China) | Schizophrenia | • Lay health workers selected from family or community coordinate care with village doctors and helped monitor early signs of relapses, facilitate linkage to primary healthcare, supervised patient medication adherence and monitored side effects<br>• Village doctors provide regular medical services and psychotropic medication management<br>• Psychiatrists provide consultation | • Pharmacotherapy<br>• Psychoeducation | • 14-item checklist for early signs of relapse and medication side effects | • Psychiatrists travel every 2 months to each township health center to provide consultations, free medication, and supervision<br>• Psychiatrists are available for consultation if severe symptoms or relapse detected | • Not described | • An e-platform delivered two daily messages to patients with psychoeducation and treatment reminders<br>• Lay health supporters and notified if 14-item checklist delivered on e-platform was positive to prompt follow-up phone call<br>• The e-platform also integrated lay health supporters work with village doctors, project coordinators and psychiatrist if severe symptoms or signs of relapse were detected |

Abbreviations: AUD, alcohol use disorder; AUDIT, Alcohol Use Disorders Identification Test; CBR, community-based rehabilitation; CBT, cognitive behavioral therapy; CHEW, community health extension workers; CHW, community health workers; EPDS, Edinburgh Postnatal Depression Scale; GHQ-28, General Health Questionnaire 28 items; GRIMS, Golombok Rust Inventory of Marital State; LHC, lay health counselors; MHFA, mental health first aid; mhGAP, Mental Health Gap Programme; MINI, Mini International Neuropsychiatric Interview; PCP, primary care physician; PHQ2, Patient Health Questionnaire 2 item; PHQ9, Patient Health Questionnaire 9 item; PST, problem-solving therapy.

**Table 3.** Summary of frequencies of Collaborative Care model components

|  | Model Component | Total (n= 20) |
|---|---|---|
| Population Management | Mental Health Screening | 18 |
|  | Registry | 2 |
| Measurement Based Care | Measurement Based Care | 11 |
|  | Treatment to Target | 8 |
| Team Roles | Care Manager | 20 |
|  | Nurse | 8 |
|  | CHW | 11 |
|  | Other | 4 |
|  | Psychiatric Consultant | 13 |
|  | Primary Medical Provider | 20 |
| Evidence Based Treatment | Pharmacotherapy | 20 |
|  | Brief Behavioral Interventions | 16 |
|  | Psychoeducation | 20 |
| Caseload Review | Systematic Caseload Review | 10 |
| Other components | Mobile/digital platform support for patient engagement | 3 |

Abbreviations: CHW, community health worker

The review extends the literature on Collaborative Care and supports its adaptability for a broad range of disorders and its dissemination to diverse settings in LMICs. The identification of components that are shared across effective models advances our understanding of what may be essential for successful implementation, which is useful information for policy makers in planning to scale Collaborative Care (Castro et al., 2020). The review also provides pragmatic information about alternative strategies for operationalizing model components based on local resources. Several of the included studies highlight the resources and planning that are required to translate a Collaborative Care model from research studies into care provided in routine clinical settings. It is not trivial, for example, to repurpose existing clinical staff to be Collaborative Care team members, but such role expansion could increase the potential sustainability of a Collaborative Care program (Stockton et al., 2020). Similarly, mental health outcome tracking could be more efficient if it is integrated into the general medical information system and workflows (Ndetei and Jenkins, 2009).

Process evaluations of several of the RCTs provide valuable insights into potential barriers and facilitators to the implementation of these Collaborative Care models. Insight into processes that work well within specific contexts can lead to increased uptake of Collaborative Care models and their capacity to address the mental health treatment gap in LMICs. Adequate training and supervision of team members are essential to facilitate implementation (Kemp et al., 2021), including training to work as a team and fostering a shared vision of the work (Li et al., 2020). Second, new tasks must be appropriate to the skills of team members and easily integrated into their existing practices. Studies that incorporated screening by clinical staff as part of routine care (rather than by research staff) highlight that this critical first step must be successful in order to achieve improved population-level outcomes – that is, that even very effective models will have limited impact if they reach very few

people who need treatment (Shidhaye et al., 2019). A 2020 review of barriers and facilitators to integrated care in LMICs highlights additional health system challenges that critically impact implementation, such as scarcity of strong leadership, lack of leadership buy-in, and mismanaged information systems (Esponda et al., 2020).

Several strengths of this rapid review increase the impact of the findings. Rigorous methods were utilized, following guidance from the Cochrane Rapid Reviews Methods Group. In addition, the review included studies related to the adult mhGAP priority mental disorders and also studies in outpatient settings beyond primary care, thus increasing the generalizability of the findings. Some limitations of this review should also be acknowledged. First, the presence of multiple components in effective models does not provide information about which (or whether) specific components are required for effectiveness. Second, because multiple disorders and heterogeneous outcomes were included, we were unable to provide a meta-analysis of the effectiveness of models. Third, we excluded cohort studies that reported outcomes but did not involve a comparison group because positive results in studies that report outcomes only for patients receiving treatment might reflect the natural course of illness, or "regression to the mean." This criterion resulted in the exclusion of several recent cohort studies of robust models that utilized a rigorous implementation approach informed by implementation science frameworks. Studies like these may provide compelling evidence for effectiveness of the model in routine clinical settings (Rimal et al., 2021).

In summary, the findings of this rapid review have important clinical implications, as they support the feasibility and effectiveness of Collaborative Care across diverse settings in LMICs. Collaborative Care provides a framework for a team to provide effective population-based, evidence-based, and measurement-based care for a range of mental disorders. The review also highlights areas where further research is needed. For effective measurement-based care, there is a need for validation and adoption of (preferably self-administered) clinical rating scales that are appropriate for different populations and different levels of the health care system (Ndetei and Jenkins, 2009; Hanlon et al., 2016). There is also a need for future studies to evaluate longer-term outcomes and to inform strategies to address implementation challenges. Randomized trial designs are poorly suited to understanding implementation barriers that are specific to a local context. Nonexperimental approaches can be used to rigorously evaluate strategies that emerge from within health systems or communities, and there is a need for more research that is informed by implementation science (McGinty and Eisenberg, 2022). In addition to lighting the path for future implementation science research about Collaborative Care in LMICs, the findings of this review can also assist health care administrators and policy makers in more effectively designing and implementing Collaborative Care models that meet their populations' integrated care needs.

## Conclusion

This rapid review provides evidence that Collaborative Care is a robust strategy to address the mental health treatment gap for common mental disorders, unhealthy alcohol use and psychosis in LMICs. Despite the more limited resources available in LMICs, effective Collaborative Care models in these settings were based on the same core principles of effective Collaborative Care in HIC settings (team-based care, population approach, evidence-based

treatments, and systematic measurement of outcomes over time to inform treatment intensification). Models operationalized these components differently, demonstrating significant innovation in tailoring to local contexts; and there was clear evidence that specific resources and support for implementation is required. These studies suggest that there is no "optimal" Collaborative Care model for all contexts. Instead, implementers and policy makers should seek the best model that is useful for a given setting (Wyrick et al., 2014), with careful consideration of affordability, efficiency and potential scalability.

**Open peer review.** To view the open peer review materials for this article, please visit http://doi.org/10.1017/gmh.2022.60.

**Supplementary material.** To view supplementary material for this article, please visit https://doi.org/10.1017/gmh.2022.60.

**Acknowledgement.** We thank Diane Powers for insights during development of this project.

**Author contributions.** T.J. collaborated in the research of rapid review methodology, designing the search strategy and completing the literature search. J.W., S.O., A.B., B.F. and L.C. outlined the objectives and purpose of this review, reviewed the search strategy, collaborated on inclusion and exclusion criteria, reviewed the final study protocol, engaged in title and abstract screening and contributed to revisions of manuscript drafts. J.W., S.O., B.F. and L.C. completed full-text reviews. J.W., S.O. and L.C. collaborated on data extraction and risk of bias assessments. J.W. and L.C. reviewed all excluded studies, reached consensus final studies included, constructed tables and collaborated on drafting the manuscript.

**Financial support.** This research received no specific grant from any funding agency, commercial or not-for-profit sectors.

**Competing interest.** The authors declare none.

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
