## [Reviewer Report]

Dear Dr. Petersen and the Editorial Board and Staff at Global Mental Health,

Thank you for the invitation to submit this manuscript titled "Successful Ingredients of Effective Collaborative Care Programs in Low- and Middle-Income Countries: A Rapid Review" as a Review Article for potential publication in the Global Mental Health.

This manuscript synthesizes the evidence for Collaborative Care in low- and middle-income countries (LMICs) and distills shared model components of effective programs by offering a rapid review of clinical trials to date regarding the effectiveness and intervention characteristics of Collaborative Care models in treating a range of psychiatric conditions in primary care and outpatient medical settings. Results from this review can help provide increased understanding of successful Collaborative Care models—including specific model components— that is vital for policy makers and healthcare systems which seek to implement Collaborative Care in LMICs. 

All authors have contributed to the writing and editing of this manuscript. None of the authors have competing interests related to this manuscript. All authors have given approval for the submission of this manuscript. This manuscript is not being considered for publication elsewhere. Results included in this manuscript have not been previously presented. This study was not funded by any sources or organizations.

Sincerely, 

Dr. Jessica Whitfield

Dr. Lydia Chwastiak

Department of Psychiatry and Behavioral Sciences

University of Washington School of Medicine

Seattle, WA, USA

---

## [Reviewer Report]

*Comments to Author*: The paper reviews collaborative care models for mental healthcare in low- and middle-income countries. This is a robust review which is very well-reported. There are some gaps in the reporting and plugging those who enhance the manuscript. One section that would greatly benefit from revision is the discussion. Further details of my suggestions are provided below.

ABSTRACT

Add a sentence on how the data was analysed

INTRODUCTION

-Provide a definition of collaborative care earlier in the manuscript to orient the readers right at the outset.

-Specify how your review is different from Cubillos et al 2021

METHODS

-Specify here that you conducted a rapid review and also why you preferred it over a systematic review

-Please provide a discoverable link for the OSF registration of protocol

-Why was the protocol not published on a peer reviewed platform designed specifically for registering review protocols such as Prospero?

-Since the focus was on LMICS, why were relevant databases such as LILACS and AJOL not searched?

-Please correct 'PsychInfo' to 'PsycInfo'

-Pleas provide details of how the data was analysed/synthesised

DISCUSSION

-It would be useful to describe an ideal collaborative care program based on your findings

-The discussion is limited to summarising the results again. There is no discussion on how this evidence from LMIC is different from or similar to evidence from HICs

-You have only specified the limitations. What about the strengths of your review?

-What are the implications of your study - clinical, research, policy?

PRISMA FLOW CHART

-Please replace 'wrong' with 'ineligible'. I believe the 'wrong' is a direct output of Covidence but is not an accurate representation of the reason for exclusion.

---

## [Reviewer Report]

*Comments to Author*: The authors have reviewed 20 Collaborative Care models in 9 LMICs and revealed that multi-disciplinary care team with structured communication, standardized protocols, well described and systematic identification of mental disorders, and a stepped-care approach to treatment are the key